# Clinical and Serological Findings of COVID-19 Participants in the Region of Makkah, Saudi Arabia

**DOI:** 10.3390/diagnostics12071725

**Published:** 2022-07-15

**Authors:** Othman R. Alzahrani, Abdullah D. Alanazi, Lauri Kareinen, Yousef M. Hawsawi, Hani A. Alhadrami, Asim A. Khogeer, Hanan E. Alatwi, Amnah A. Alharbi, Tarja Sironen, Olli Vapalahti, Jussi Hepojoki, Fathiah Zakham

**Affiliations:** 1Department of Biology, Faculty of Science, University of Tabuk, Tabuk 71491, Saudi Arabia; o-alzahrani@ut.edu.sa (O.R.A.); h_alatwi@ut.edu.sa (H.E.A.); 2Genome and Biotechnology Unit, Faculty of Science, University of Tabuk, Tabuk 71491, Saudi Arabia; ahalharbi@ut.edu.sa; 3Department of Biological Sciences, Faculty of Science and Humanities, Shaqra University, P.O. Box 1040, Ad-Dawadimi 11911, Saudi Arabia; aalanazi@su.edu.sa; 4Department of Virology, Medicum, Faculty of Medicine, University of Helsinki, 00014 Helsinki, Finland; lauri.kareinen@helsinki.fi (L.K.); tarja.sironen@helsinki.fi (T.S.); olli.vapalahti@helsinki.fi (O.V.); jussi.hepojoki@helsinki.fi (J.H.); 5Department of Veterinary Biosciences, Faculty of Veterinary Medicine, University of Helsinki, 00014 Helsinki, Finland; 6Research Center, King Faisal Specialist Hospital and Research Center, P.O. Box 40047, Jeddah 21499, Saudi Arabia; hyousef@kfshrc.edu.sa; 7College of Medicine, Al-Faisal University, P.O. Box 50927, Riyadh 11533, Saudi Arabia; 8Department of Medical Laboratory Technology, Faculty of Applied Medical Sciences, King Abdulaziz University, P.O. Box 80402, Jeddah 21589, Saudi Arabia; hanialhadrami@kau.edu.sa; 9Special Infectious Agent Unit, King Fahd Medical Research Centre, King Abdulaziz University, P.O. Box 80402, Jeddah 21589, Saudi Arabia; 10Plan and Research Department, General Directorate of Health Affairs Makkah Region, MOH, Mecca 24321, Saudi Arabia; akhogeer@moh.gov.sa; 11Department of Biochemistry, Faculty of Sciences, University of Tabuk, Tabuk 71491, Saudi Arabia; 12HUS Diagnostic Center, HUSLAB, Clinical Microbiology, Helsinki University Hospital, University of Helsinki, 00014 Helsinki, Finland; 13Institute of Veterinary Pathology, Vetsuisse Faculty, University of Zürich, 8057 Zürich, Switzerland; 14Faculty of Pharmacy, University of Helsinki, 00014 Helsinki, Finland

**Keywords:** SARS-CoV-2, ELISA, micro-neutralization assay, IgM, IgA, IgG ELISA, Makkah, Saudi Arabia

## Abstract

Makkah in Saudi Arabia hosts the largest annual religious event in the world. Despite the many strict rules enacted, including Hajj cancellation, city lockdowns, and social distancing, the region has the second highest number of new COVID-19 cases in Saudi Arabia. Public health interventions that identify, isolate, and manage new cases could slow the infection rate. While RT-PCR is the current gold standard in SARS-CoV-2 identification, it yields false positive and negative results, which mandates the use of complementary serological tests. Here, we report the utility of serological assays during the acute phase of individuals with moderate and severe clinical manifestations of SARS-CoV-2 (COVID19). Fifty participants with positive RT-PCR results for SARS-CoV-2 were enrolled in this study. Following RT-PCR diagnosis, serum samples from the same participants were analyzed using in-house ELISA (IgM, IgA, and IgG) and microneutralization test (MNT) for the presence of antibodies. Of the 50 individuals analyzed, 43 (86%) showed a neutralizing antibody titer of ≥20. Univariate analysis with neutralizing antibodies as a dependent variable and the degree of disease severity and underlying medical conditions as fixed factors revealed that patients with no previous history of non-communicable diseases and moderate clinical manifestation had the strongest neutralizing antibody response “Mean: 561.11”. Participants with severe symptoms and other underlying disorders, including deceased individuals, demonstrated the lowest neutralizing antibody response. Anti-spike protein antibody responses, as measured by ELISA, showed a statistically significant correlation with neutralizing antibodies. This reinforces the speculation that serological assays complement molecular testing for diagnostics; however, patients’ previous medical history (anamnesis) should be considered in interpreting serological results.

## 1. Introduction

Coronavirus disease 2019 (COVID-19) was initially identified in December 2019, in the city of Wuhan, located in the Hubei province of China [1,2]. On 30 January 2020, the World Health Organization (WHO) declared COVID-19 as a Public Health Emergency of International Concern (PHEIC) and eventually a pandemic.

People from more than 180 different countries come to Saudi Arabia, as it hosts the largest mass gathering in the world during pilgrimage and Umara in Makkah. Additionally, the country has global trade relationships with China [3].

On 27 February, the Saudi Arabian government suspended entry to Makkah and Medina, where most of the religious rituals take place, to restrict mass gatherings [4]. On 2 March 2020, a traveler arriving from abroad was confirmed as the first Saudi Arabian COVID-19 case. Several additional cases were reported around the same period. As a result, the health authorities in Saudi Arabia decided to take an action to prevent the rapid spread of the disease [5].

Social distancing control measures were also enforced with a country-wide lockdown to reduce contact between people and to interrupt the transmission chains. In addition, most flights were temporarily re-suspended. By the end of July, the ministry of pilgrimage affairs allowed the pilgrimage only to the people residing in Saudi Arabia (including foreigners). This caused the number of pilgrims to be reduced from more than 3 million to only a few thousand [6]. Further, the pre-selection of pilgrims was based on a special quota system, and strict rules were adopted by the ministry of health (MOH) during the religious rituals. These rules included a safety bubbling strategy to avoid mass gatherings and limit transmission of COVID-19 [6].

Despite all the efforts to contain the spread, the Saudi Ministry of Health documented 765,788 confirmed cases and 9140 deaths in all Saudi regions on 27 May 2022 [7]. In addition, Saudi Arabia has also been affected by Middle East respiratory syndrome (MERS), which has been known since April 2012 [3]. According to the latest WHO reports (1 April 2020–31 May 2020), the National IHR Focal Point of the kingdom documented nine new MERS-CoV cases, including five deaths. Further, six patients, including a health care professional, were reported in an outbreak at the hospitals in the Riyadh region [8]. Due to the novelty of COVID-19 and its high contagiousness (similar to influenza), limited options were available to control its spread and to manage cases. The diagnosis of SARS-CoV-2 is currently based on the detection of viral RNA in nasopharyngeal swabs [9], as well as antigen tests to detect certain viral proteins [10,11].

Reverse transcriptase polymerase chain reaction (RT-PCR) based assays have been considered the gold standard for detecting of SARS-CoV-2; however, various limitations are associated with their accuracy, including false negative results in up to 30% of cases [12,13]. These are mainly due to the precariousness of material availability and the change in accuracy over the course of the disease. In addition, the success of RT-PCR based diagnosis depends primarily on the pre-analytic phase of testing and the quality of the nasal or nasopharyngeal swabs. Ensuring that the pre-analytic phase and quality of materials are properly handled is challenging when handling such a vast number of patients in hospital settings. While there are issues in clinical sensitivity, particularly in cases of delayed access to diagnostics, as well as issues in addressing prolonged viral RNA shedding, RT-PCR remain the diagnostic method of choice for an acute COVID-19 diagnosis. Though there is some debate on the convenience and accuracy of the use of RT-PCR in the decision-making process on infection clearance and control of transmission [14], serology has its use particularly in retrospective evaluation of previous infection or immunization and population immunity studies.

On the other hand, this has shed a light on the benefits of serology, which could serve as a complementary test for the confirmation of SARS-CoV-2 exposure, in particular in cases featuring the clinical presentation of COVID-19, but a negative RT-PCR result. Serology also enables prevalence and accumulative incidence measurements, as well as case monitoring for epidemiological and surveillance studies [15,16]. The enzyme-linked immunosorbent assay (ELISA) is one of the most commonly applied techniques for detecting antibodies in COVID-19 patients [16]. Serological assays for COVID-19 diagnosis are based on recombinant antigens, mainly the immune-dominant spike protein (the immuno-dominant protein) of SARS-CoV-2, and in general show high sensitivity and specificity, as depicted in this study [17].

Microneutralization tests (MNT), on the other hand, measure neutralizing antibodies in a patient sample [18]. However, MNT with infectious SARS-CoV-2 requires a biosafety level 3 (BSL3) facility and experienced staff to handle the virus, which limits its use in routine analysis.

In this study, we evaluated the performance of SARS-CoV-2 spike protein ELISA assay with MNT for the detection and early diagnosis of COVID-19 in patients with RT-PCR confirmed SARS-CoV-2 infections. We also compared the level of SARS-CoV-2 neutralizing antibodies in participants with moderate or severe clinical forms of disease and underlying conditions in the region of Makkah in Saudi Arabia.

## 2. Materials and Methods

### 2.1. Study Design and Ethical Approval

In the current study, we enrolled 50 individuals diagnosed with COVID-19 who had fever and respiratory infection symptoms and were admitted to the East Arafat Hospital From 1 to 13 in June 2020 (before the emergence of variants of concern), in the Makkah region of Saudi Arabia. Informed consent forms were signed by all the patients. Nasopharyngeal swabs from these participants were tested for SARS-CoV-2 using the PowerCheckTM 2019-nCoV Real time PCR (kit: R6900TD) and LightCycler480 instrument II. This study was approved by the Institutional Review Board of Ministry of Health in Saudi Arabia (IRB number H-02-K-076-00520-298).

### 2.2. Sample Collection

A total of 3 to 5 mL blood sample was collected from each confirmed COVID-19 individuals in VACUETTE^®^ Blood Tube containers without adding any anticoagulant or preservative. The demographic data (age, sex, race, health and social status, region) of participants diagnosed clinically with COVID-19 were retrieved according to the case definition of the MOH in Saudi Arabia. All the procedures were performed according to the recommendation of the WHO and institutional protocols. Serum samples were collected in the acute phase between 0–5 days of admission to hospital, and most patients were hospitalized 7 days after the onset of symptoms. Serum from each sample was separated from cells within one hour after the blood collection. The collected samples were centrifuged at 15,115× *g* for 5 min. The sera were collected in new tubes and inactivated at 56 °C for 30 min, then frozen at −80 °C until use. The serum inactivation step was performed at the research center of King Faisal Specialist Hospital and Research Center, (KFSHRC), Jeddah in Saudi Arabia. The sera samples were shipped to the University of Helsinki in Finland for further analysis.

### 2.3. Serological Assays

#### 2.3.1. Microneutralization Test (MNT)

MNT for neutralizing SARS-CoV-2 antibodies, as described in [19], was carried out at the BSL-3 facility at the University of Helsinki. In brief, Vero E6 cells were grown on a 96-well plate in Eagle’s Minimum Essential Medium (EMEM), supplemented with 7.5% Fetal Bovine Serum (FBS) and antibiotics 100 IU penicillin, 100 ug/mL streptomycin, and 2 mM L-glutamine. They were then incubated in +37 °C and 5% CO_2_ until approximately 90% confluent.

The sera were serially diluted (1:20, 1:40, 1:80, 1:160, 1:320, 1:640, and 1:1280). A volume of 50 µL from each dilution was mixed with 50 µL of diluted virus stock (approximately 50 pfu/well) and incubated for 1 h at +37 °C. The 96-well plate was inoculated with the serum-virus-mix and kept for 4 days at +37 °C with 5% CO_2_. Then the cells on the plate were fixed, stained with crystal violet, and the titers were read based on cytopathic effect (CPE). As the samples were collected in the acute phase and all the patients were having COVID-19 symptoms and were RT-PCR positive for SARS-CoV-2, a titer of 20 neutralizing antibodies or more was considered to be positive.

#### 2.3.2. Enzyme-Linked Immunosorbent Assay (ELISA) for SARS-CoV-2 S Protein

The recombinant antigens were produced, and the assay was set up initially following a described method [20], but was then modified as described by Rusanen et al. [21]. The assay was used to determine presence of IgM, IgA, and IgG class antibodies against SARS-CoV-2 S protein and have been evaluated to have a sensitivity of 100% in detecting IgG seroconversion more than 13 days after onset of illness in SARS_COV-2 RT-PCR positive COVID-19 patients.

### 2.4. Statistical Analysis

Descriptive statistics were performed by the use of SPSS for windows version 23, IBM Corp. (New York, NY, USA).

The univariate analysis (two-way analysis of variance) was done to assess the differences between the neutralizing antibody titers, the severity of the diseases according to clinical status, and other underlying medical conditions. Correlation tests were performed using Pearson’s test to determine the correlation between the production of neutralizing antibodies and different immunoglobulins in the acute phase of infection. The correlation between SARS-CoV2-specific antibodies was measured with neutralization and ELISA tests. Prism Graphpad version 9 was used for Pearson’s test and bubble plotting.

## 3. Results

### 3.1. Clinical and Demographic Data of Patients

A total of 50 participants were enrolled in this study after RT-PCR confirmation. The age of patients ranged from 19 to 90, with a median age of 52.5 years. Most individuals were males (*n* = 44, 88.0%), and 5 patients (age above 54) died within a few days of diagnosis. Only one patient, an individual who had a history of cardiovascular disease and hypertension, died at the age of 30. The rest of the deceased patients (*n* = 4, 80%) had a history of cardiovascular problems, and one had cancer. Twenty-seven patients were reported with one or more underlying diseases (*n* = 27, 54.0%).

Most of patients suffered from a moderate form of COVID-19 (*n* = 32, 64%), whereas the rest (*n* = 18, 36.0%) of the patients demonstrated a severe form of the disease and underwent intubation with intensive care program. Notably, most patients (*n* = 37, 74%) were non-Saudi, coming mainly from Asia (Pakistan, Bangladesh, India, Myanmar, Indonesia, and Yemen) and Africa (Egypt, Mali, Nigeria, and Ethiopia). Table 1 summarizes the demographic and clinical data of the patients.

### 3.2. MNT Results

The MNT assay was performed on samples collected during the acute phase of illness (50 serum samples), and 43 (86%) of patients showed neutralizing antibody titers of ≥20. Univariate analysis was performed, considering the neutralizing antibody titers as a dependent variable, and the degree of severity and underlying medical conditions as fixed factors. The results showed that patients with no previous history of non-communicable diseases and who presented only moderate clinical manifestation had the highest mean titer (mean = 561.11) of neutralizing antibodies. The patients with severe symptoms and other underlying diseases showed a lower amount of neutralizing antibodies (Figure 1). Eighteen patients (*n* = 18, 36%) with severe symptoms, including five deceased individuals, displayed a lower mean titer (mean = 94.44) of neutralizing antibodies. The samples were collected between days 1 and 5 after admission with clear symptoms, partially explaining the observed difference in the antibody responses. Three of the participants who did not survive showed neutralizing antibodies of less than 20, and two of the deceased participants showed a neutralizing antibody titer of 160.

### 3.3. ELISA IgM, IgA, and IgG Assays Results

We compared the ELISA and MNT results using the Pearson correlation test, and found a statistically significant positive correlation between IgM, IgA, and IgG antibody levels and MNT (*r* = 0.47, *r* = 0.37 and *r* = 0.42, *p* values 0.001, 0.009 and 0.003) as shown in Figure 2. Further, plotting the results of neutralizing antibodies versus IgM antibodies, degree of severity, and underlying diseases showed that most of the individuals with high antibodies developed moderate manifestations. Only one patient with severe symptoms and not suffering from underlying diseases showed higher antibody values for IgM and MNT (Figure 3).

## 4. Discussion

The severity of COVID-19 varies greatly, and has been considered the most serious pandemic to mankind since the Spanish flu pandemic in 1918 [22]. Global dissemination has been rapid, and the number of newly detected infections continues to increase rapidly worldwide. Further, COVID-19 spreads efficiently, meaning countries hosting religious and sporting mass gathering events face enormous challenges to evaluate and mitigate the risk of disease transmission.

Saudi Arabia hosts the largest planned and recurrent annual mass gathering (Hajj) at a specific lunar time, as well as spontaneous small pilgrimages (Umrah) which can take place at any time of the year. Both events are associated with the risk of transmission of various infectious diseases. Most pilgrims develop clinical symptoms associated with various pathogens of the respiratory tract [23]. Regular, continuous, timely, and accurate diagnosis is among the operational, preventive, and relevant responses to contain and mitigate the impact engendered by disease during religious events. The Saudi Arabian health authorities employ various preventive measures against infectious diseases prior to and during the pilgrimage. For instance, the Hajj in year 2021 was cancelled for all people coming from abroad, and only a small number of residents were able to perform the pilgrimage. These residents underwent RT-PCR testing before and after the event to confirm that they were SARS-CoV-2 free.

As hospitalized and diagnosed cases comprise only a fraction of the infections, serological testing allows monitoring the immunity to COVID-19 within the population, which is important for decision-making on control efforts. It also provides individual-level information on susceptibility and the ability to work with these vulnerable individuals (e.g., risk groups without infection risk).

Typically, the patients became antibody positive during the second week of illness; however, most of them developed measurable antibodies to SARS-CoV-2 by the third week of infection. This explains the moderate correlation between the neutralized antibodies and IgM, IgA, and IgG antibodies in our study. Furthermore, seroconversion was found to occur within two weeks of symptom onset [24]. Another study showed that IgM antibody levels were high only in symptomatic patients, while IgG seroconversion occurred in most COVID-19 asymptomatic and symptomatic infected people [25]. The present study is in agreement with other studies, as IgM antibodies were present in samples collected between the first and fifth days of sampling [26] and IgG antibodies could also be detected in the same samples.

A large-scale study revealed that more than a third of non-RT-PCR-confirmed SARS-CoV-2 patients showed IgG seropositivity after recovery in New York City [14]. The study also showed that IgG were developed during a period of 7–50 days from symptom onset. We reported here that around 80% of the participants included formed IgG antibodies during the acute phase or between the time of admission to hospital and sample collection.

These findings also affirm that antibody level may vary between individuals at different phases of infection. Recently, Norman et al. developed a multiplex fluorescence-based assay for the early detection of seroconversion at the early phase of symptom onset in parallel with the first positive RT-PCR [27]. Our study contradicts the finding of reports showing that neutralizing antibodies positively correlate with severity in the convalescent phases or after recovery in opposite to our results [26,27]. However, these studies did not provide information about the serology of cases during the term of infection (acute phase), and the time interval between the acute and convalescent phases should be considered for comparison [28,29]. Further, comorbidity, or the presence of other underlying medical conditions, could also be a factor exacerbating the patient’s clinical status regardless of neutralizing antibody levels. A recent cohort study in the UK confirmed that patients who survived SARS-CoV-2 infection showed to have more neutralizing antibodies compared to deceased individuals from the disease. Similarly in our study, deceased people suffering from other clinical manifestations also showed lower neutralizing and other measurable antibody release when compared to the participants who have underlying diseases with moderate manifestations [30]. Further, several studies have showed that cardiac vascular dysfunctions and metabolic diseases could cause humoral immunity disturbance and affect the production of antibodies [31,32,33].

## 5. Study Limitations

The sample size is small, and the participants included in this study had either moderate or severe symptoms. No asymptomatic individuals with mild symptoms were available to be included for comparison because the target group, i.e., patients admitted to the hospital, have clear symptoms. Participants were in the acute phase of infection and the time of symptom onset was unclear in some cases. The study was done on samples collected in the acute phase and not the convalescent phase. Moreover, the recovered patients were only tested by RT-PCR, showing negative positive results [14].

## 6. Conclusions

This study addressing the overall and neutralizing antibody response against SARS-CoV-2 among individuals hospitalized due to COVID-19 in Saudi Arabia demonstrates that robust immune responses develop in most participants. However, individuals with very severe or fatal infections with comorbidities showed reduced neutralizing antibody levels. The results suggest that neutralizing antibody or antibody levels in general may differ based on the patient’s clinical condition. Serological assays can complement nucleic acid and antigen detection tests by confirming recent SARS-CoV-2 infection or vaccination status. However, considering the limitations of this study, it is difficult to generalize the findings.

Further, nationwide seroprevalence and epidemiological studies are needed to investigate protective or neutralizing immunity and exposure levels.

## Figures and Tables

**Figure 1 diagnostics-12-01725-f001:**
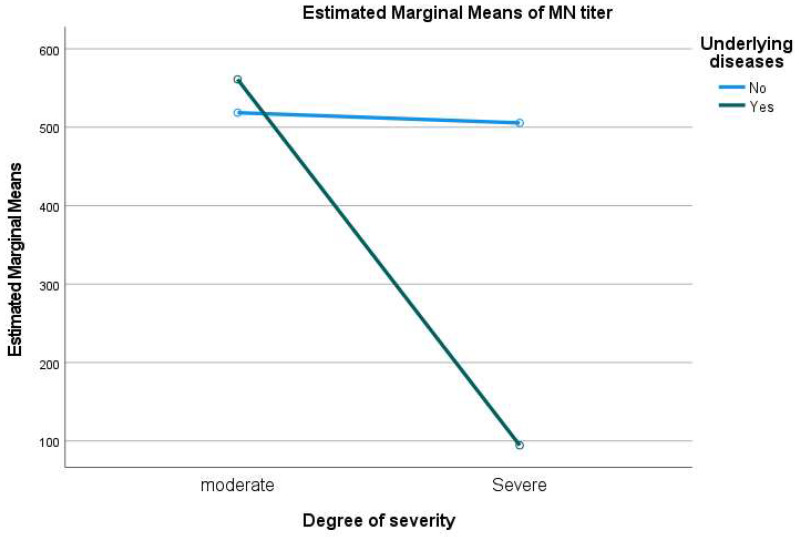
Univariate analysis MN titers versus degree of severity and underlying diseases.

**Figure 2 diagnostics-12-01725-f002:**
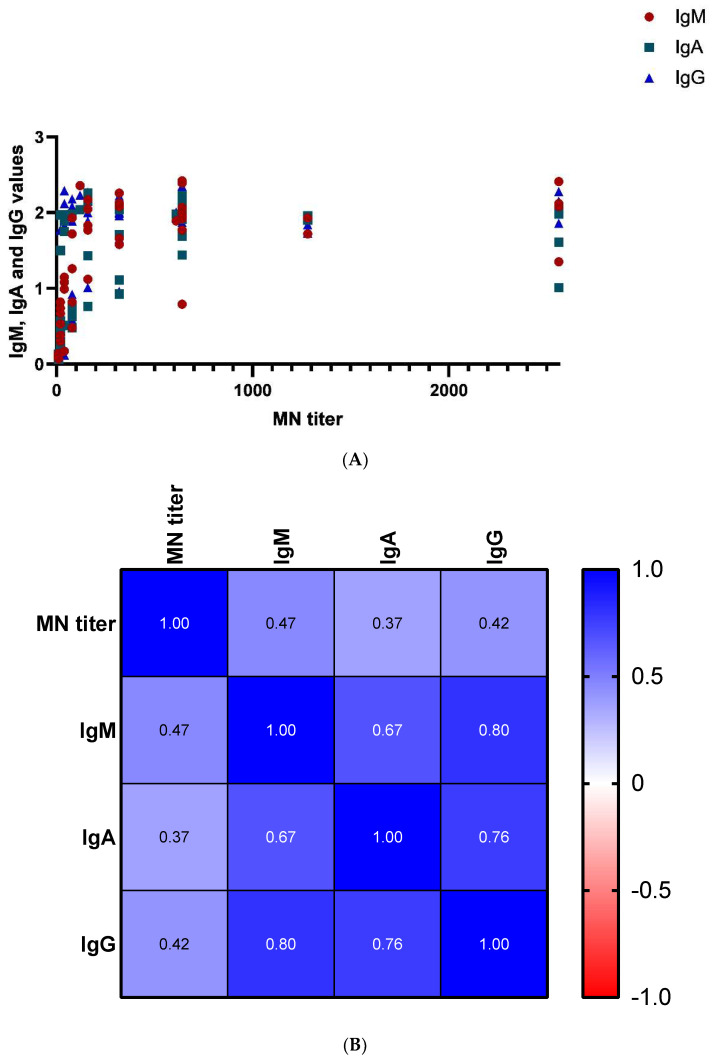
(**A**) Plotting of MN titers versus ELISA IgM, IgA, and IgG values. (**B**) Pearson’s test matrix: *p* values 0.001, 0.009, and 0.003, respectively.

**Figure 3 diagnostics-12-01725-f003:**
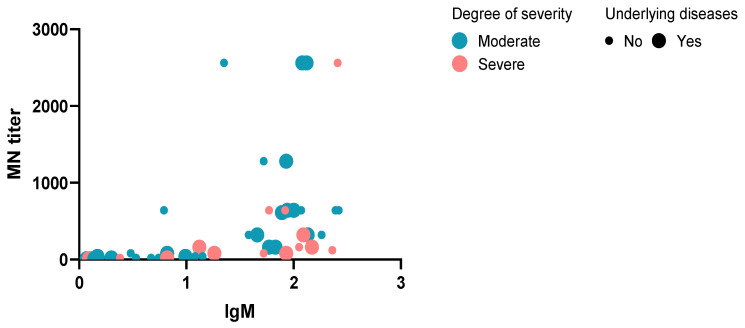
Bubble plot MN titers versus IgM antibodies according to the degree of severity and underlying diseases status.

**Table 1 diagnostics-12-01725-t001:** Demographic and clinical characteristics of participants in Makkah.

Characteristics	Number and Percentage
Demographic data	
Age range	
≤20	2 (4.0%)
21–30	4 (8.0%)
31–40	9 (18.0%)
41–50	8 (16.0%)
51–60	11 (22.0%)
≥61	16 (32.0%)
Sex	
Male	44 (88.0%)
Female	6 (12.0%)
Origin	
Saudi	13 (26.0%)
non-Saudi	37 (74.0%)
Smoking	
Smoker	22 (46.0%)
Nonsmoker	27 (44.0%)
Not known	1 (2.0%)
Outcome	
Died	5 (10.0%)
Survived	45 (90.0%)
Other underlying medical conditions	
Autoimmune diseases	1 (2.0%)
Liver diseases	2 (4.0%)
Malignancy	1 (2.0%)
Kidney diseases	3 (6.0%)
Cerbro-vascular diseases	5 (10.0%)
Lung diseases	3 (6.0%)
Cardiovascular diseases	14 (28.0%)
Diabetes mellitus	12 (24.0%)
Arterial Systematic Hypertension	25 (50.0%)
Clinical manifestations	
Fever (>38.0 °C)	10 (20%)
Headache	6 (12%)
GIT symptoms (nausea, vomiting, diarrhea)	7 (14%)
Sore throat	6 (12%)
Nasal congestion	6 (12%)
Runny nose	2 (4%)
Cough (dry or productive)	28 (56%)
Hemoptysis	5 (10%)
Shortness of breath	23 (46%)
Conjunctival congestion	2 (4%)
Intubated individuals	18 (36%)

## Data Availability

Data are available in the main text.

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
