# Peer review of "Clinical and Serological Findings of COVID-19 Participants in the Region of Makkah, Saudi Arabia"

_diagnostics, 2022, doi:10.3390/diagnostics12071725_

Round 1
Reviewer 1 Report
The comments from my first run peer review were satisfactory answered and implemented into the text of the paper. Therefore, I recommend to accept revised paper for publication in the Diagnostics.
Author Response
We’re glad to hear that the reviewer is satisfied about the modified version and we appreciate all the valuable comments that helped us to improve the quality of the manuscript.
Reviewer 2 Report
Clinical and Serological findings of COVID-19 patients in the region of Makkah, Saudi Arabia is a well-structured study, methodologically adequate to answer the objective question: what is the usefulness of serology in the acute phase of moderate-severe infection by SARS-COV-2?. The limitations are clearly related to the small sample size, but the comparisons are correct. Maybe the number of patients with a severe clinical presentation is too low. Methods are presented with clarity and reproducibility, and the results are expressed in the order of the study. Furthermore, they are supported by very timely and well explained Tables and Figures. It would be advisable to update some bibliographical references.
Author Response
We thank the referee for his comments and suggestions. As we mentioned in the first round of peer reviewing, SARS-CoV 2 serology is a complementary testing to PCR for a diagnosis purpose, but it is important to measure the prevalence and tracking the spread of infection in the population over the time, especially in the case of Makkah. We agree about the low number of included subjects and we highlighted this point in the study limitation. We have also added and updated the references list.
Reviewer 3 Report
Article: Clinical and Serological findings of COVID-19 patients in the region of Makkah, Saudi Arabia
Study presentation
In this study, it was evaluated the performance of SARS-CoV-2 spike protein ELISA assay with MNT for the detection and early diagnosis of COVID-19 in individuals with RT-PCR confirmed SARS-CoV-2 infection. It was also compared the level of SARS-CoV-2 neutralizing antibodies in individuals with moderate or severe clinical forms of disease and underlying conditions in the region of Makkah in Saudi Arabia.
Major comments
The study was two main limitations: (i) a low number of individuals enrolled; and (ii) the study was restricted to only a region decreasing the generality of its findings.
It is also important to change the term “patients” to “participants or individuals”
The authors included only patients with moderate and severe phenotypes. In this context, it is difficult to analyze the results. Also, the statistical approach should be revised.
Abstract
To correct the sentence “… new cases could slow the infection rate. While RT-PCR is the current gold standard in SARS-CoV-2 diagnostics.” To “… new cases could slow the infection rate. While RT-PCR is the current gold standard in SARS-CoV-2 identification.” Also, to change “with moderate and severe clinical manifestations of SARS-COV-2.” To “with moderate and severe clinical manifestations of SARS-COV-2 infection (COVID-19).”
To correct “>=” to “≥”
To correct “Univariate analysis” to “Bivariate analysis”. The authors are comparing two variables.
To remove the “space” here “(Mean: 561, 11). Also to use “,”.
Introduction
The first sentence can be deleted.
To avoid the use of adjectives such as “vibrant”, “serious”
To use abbreviations when the term was used more than 3 times. This information should be used in the manuscript for all sections.
Methods
To use superscript letters to “®”
To use “race” in place of “ethnicity”
To correct “ml” to “mL”
To correct “CO2” to “CO2”
To correct “1:1280” to “1:1,280”
To correct “+37C” to “+37 oC”
The following excerpt is a result: “The specificities of IgG, IgM, and IgA tests were 100%, 100%, and 98%, respectively. Also, in materials including pre-pandemic samples, the sensitivities to detect sero-conversion for the same were 90, 66, and 91%, respectively.
To describe the period of the study (onset and final period of data collection).
To include detailed information for clinical and demographic data.
Statistical analysis
The authors should include the analysis for normality. Also, it is important to cite the software used to perform the statistical analysis and the alpha error used to decide on the hypothesis. The authors should include the descriptive approach applied in the study. In addition, the authors could include a correction by multiple tests, such as Bonferroni or false rate discovery test. Finally, it is crucial to include the power of the study.
Results
To use the number only above 10.
To correct the onset of this sentence “The other patients suffered a mild form of COVID-19 (n=32, 64%), whereas the rest (n=18, 36.0%) of the patients demonstrated severe form of the disease and underwent intubation with intensive care program.” Reading this sentence, we can conclude that the authors have 50 patients (32 mild and 18 severe) + the patients cited in the first part of the paragraph.
Table 1.
To correct “=>20” to “≤20”. Also, to correct “=<61” to “≥60”. To inform “years of age” in the Table. Also, the authors should cite a reference for its age cutoffs.
To change “genre” to “sex”
To correct “non- Saudi” to “non-Saudi”. In addition, describe the origin type, e.g., place where the individuals lived
To change the term “deaths” to “outcome”. Also, the authors could cite this data as the last one
To change “diabetes” to “diabetes mellitus”
To change “Hypertension” to “Arterial Systematic Hypertension”
Importantly, are there other conditions? If yes, the authors can include a new category “other”. The same can be done for symptoms.
Intubated patients are severe cases of the disease. However, some patients can die due to severity before being intubated. In my opinion, the authors should only cite that the patients were intubated without classifying them into severity degrees.
Important: To include the decimal cases.
Attention to the use of univariate term. The authors compared two variables, in that sense, we have a bivariate analysis.
To use only Figure 1 or Table 2. Both presented the same data. Also, it is important to include the P-values and the statistical test used to perform the study.
Figure 1. To correct “moderate” to “Moderate”. Also, in my opinion, the authors should only cite intubated and non-intubated.
Figure 2. Part A is difficult to read. Part B. The authors can delete the number above the 1. The P-values should be cited. Also, the authors should classify the correlations between IMT and IGA, IGG, and IGM as moderate, for example.
Figure 3. Is great. However, the authors should include a detailed legend.
Table 2 can be excluded.
Supplementary file 1 was not uploaded.
Discussion
No comments
Conclusion
The authors can conclude the study considering its limitations. It is difficult to generalize the findings.
Author Response
- Responses to Reviewer 3:
- Comments and Suggestions for Authors:
Study presentation:
In this study, it was evaluated the performance of SARS-CoV-2 spike protein ELISA assay with MNT for the detection and early diagnosis of COVID-19 in individuals with RT-PCR confirmed SARS-CoV-2 infection. It was also compared the level of SARS-CoV-2 neutralizing antibodies in individuals with moderate or severe clinical forms of disease and underlying conditions in the region of Makkah in Saudi Arabia.
3.1 Major comments:
The study was two main limitations: (i) a low number of individuals enrolled; and (ii) the study was restricted to only a region decreasing the generality of its findings.
It is also important to change the term “patients” to “participants or individuals”
The authors included only patients with moderate and severe phenotypes. In this context, it is difficult to analyze the results. Also, the statistical approach should be revised.
Response to comment 3.1.:
We thank the referee for these comments and suggestions. We agree with the reviewer about the low number size of study subjects and we highlighted this point as a study limitation. However, the region of Makkah has a high population comparing to other regions in the country. Further, it hosts the biggest religious mass gathering in the world. Makkah also hosts Alumrah events during the whole year and represent the most important region in term of infectious diseases dissemination. We considered the requested modification about our study subjects and the term “patients” is changed to “participants or individuals including the title.
We have checked our statistical approach with a consultant biostatistician from the University of Helsinki (Dr Ville Kinnula) and due to low sample size, Dr Kinnula agreed with the reviewer and advised us to perform only descriptive statistics and not any hypothesis based statistical testing. Consequently, we removed Anova analysis results, because we cannot draw a conclusion about population from this low sample size.
Comment 3.2: Abstract
Comment 3.2.1: To correct the sentence “… new cases could slow the infection rate. While RT-PCR is the current gold standard in SARS-CoV-2 diagnostics.” To “… new cases could slow the infection rate. While RT-PCR is the current gold standard in SARS-CoV-2 identification.”
Response 3.2.1:
Correction has been done.
Comment 3.2.2: Also, to change “with moderate and severe clinical manifestations of SARS-COV-2.” To “with moderate and severe clinical manifestations of SARS-COV-2 infection (COVID-19).”
Response 3.2.2:
Correction has been done.
Comment 3.2.3: To correct “>=” to “≥”
Response 3.2.3:
Correction has been done.
Comment 3.2.4: To correct “Univariate analysis” to “Bivariate analysis”. The authors are comparing two variables.
Response 3.2.4:
We used the univariate analysis (two-way analysis of variance) by SPSS. The dependent variable was the neutralizing antibody titers, the degree of severity, and underlying medical conditions were used as fixed factors and we clarified this point in the manuscript.
Comment 3.2.5: To remove the “space” here “(Mean: 561, 11). Also to use “,”.
Response 3.2.5: Correction has been done.
Comments 3.3 Introduction:
Comments 3.3.1: The first sentence can be deleted.
Response 3.2.1: As suggested, the first sentence has been deleted in the introduction part.
Comments 3.3.2: To avoid the use of adjectives such as “vibrant”, “serious”
Response 3.2.2: we avoided the use of the mentioned adjectives.
Comments 3.3.3: To use abbreviations when the term was used more than 3 times. This information should be used in the manuscript for all sections.
Response 3.2.3: Needful action is done in the manuscript.
Comments 3.4 Methods:
Comments 3.4.1: To use superscript letters to “®”
Response 3.4.1: This has been avoided.
Comments 3.4.2: To use “race” in place of “ethnicity”
Response 3.4.2: Correction has been done in the manuscript.
Comments 3.4.3: To correct “ml” to “mL”
Response 3.4.3: The correction has been done in the manuscript.
Comments 3.4.4: To correct “CO2” to “CO2”
Response 3.4.4: The correction has been done in the manuscript.
Comments 3.4.5: To correct “1:1280” to “1:1,280”
Response 3.4.5:.The correction has been done in the manuscript.
Comments 3.4.6: To correct “+37C” to “+37 oC”
Response 3.4.6: The correction has been done in the manuscript.
Comments 3.4.7: The following excerpt is a result: “The specificities of IgG, IgM, and IgA tests were 100%, 100%, and 98%, respectively. Also, in materials including pre-pandemic samples, the sensitivities to detect sero-conversion for the same were 90, 66, and 91%, respectively.
Response 3.4.6: Thank you very much for your suggestion .The correction has been done in the manuscript. The sentence has been deleted in the methodology part.
Comments 3.4.7: To describe the period of the study (onset and final period of data collection)
Response 3.4.7: From 01/06/2020 to 19/06/2020.
Comments 3.4.8: To include detailed information for clinical and demographic data.
Response 3.4.8: Most of the demographic and clinical characteristics of participants have been presented in table 1. We can show you also all the data but these information are confidential and cannot be published.
Comments 3.5 Statistical analysis:
Comment 3.5.1: The authors should include the analysis for normality. Also, it is important to cite the software used to perform the statistical analysis and the alpha error used to decide on the hypothesis. The authors should include the descriptive approach applied in the study. In addition, the authors could include a correction by multiple tests, such as Bonferroni or false rate discovery test. Finally, it is crucial to include the power of the study.
Response 3.5: As we mentioned, the sample size was very low and the power of study was not calculated (considering alpha or beta errors) before running the study and we performed serology testing for the samples that we got. According to our consultant, no hypothesis based statistical analysis is applicable. We accordingly removed the Anova analysis results stated previously in the manuscript and in the abstract. Descriptive statistics and Univariate analysis were performed by the use of SPSS for windows version 23, IBM Corp NY. Pearson’s correlation test was used to assess the relationship between two quantitative variables and Prism Graphpad version 9 was used for this purpose.
Comments 3.6 Results:
Comment 3.6.1: To use the number only above 10.
Response 3.6.1: this has been considered.
Comment 3.6.2: To correct the onset of this sentence “The other patients suffered a mild form of COVID-19 (n=32, 64%), whereas the rest (n=18, 36.0%) of the patients demonstrated severe form of the disease and underwent intubation with intensive care program.” Reading this sentence, we can conclude that the authors have 50 patients (32 mild and 18 severe) + the patients cited in the first part of the paragraph.
Response 3.6.2: this has been restructured accordingly in the modified manuscript.
Comments 3.7 Tables and figures:
Table 1
Comment 3.7.1: To correct “=>20” to “≤20”. Also, to correct “=<61” to “≥60”. To inform “years of age” in the Table. Also, the authors should cite a reference for its age cutoffs.
Response 3.7.1: correction has been done in the manuscript.
Comment 3.7.2: To change “genre” to “sex”
Response 3.7.2: correction has been done in the table No 1.
Comment 3.7.3: To correct “non- Saudi” to “non-Saudi”. In addition, describe the origin type, e.g., place where the individuals lived.
Response 3.7.3: correction has been done in the manuscript. We stated the origin of patients in the main text. According to the ethics committee board, the data of patients can not be shared in public and consequently we removed the supplementary file from the main manuscript. But we can share it for verification purposes.
Comment 3.7.4: To change the term “deaths” to “outcome”. Also, the authors could cite this data as the last one
Response 3.7.4: correction has been done in the manuscript.
Comment 3.7.5: To change “diabetes” to “diabetes mellitus”
Response 3.7.5: correction has been done in the manuscript.
Comment 3.7.6: To change “Hypertension” to “Arterial Systematic Hypertension”
Response 3.7.6: correction has been done in the manuscript.
Comment 3.7.7: Importantly, are there other conditions? If yes, the authors can include a new category “other”. The same can be done for symptoms.
Response 3.7.7: No, all the conditions have been included in table 1.
Comment 3.7.8: Intubated patients are severe cases of the disease. However, some patients can die due to severity before being intubated. In my opinion, the authors should only cite that the patients were intubated without classifying them into severity degrees.
Response 3.7.8: We have mentioned Intubated participants without degree of severity.
Comment 3.7.9: Important: To include the decimal cases
Response 3.7.9: this has been considered.
Comment 3.7.10: Attention to the use of univariate term. The authors compared two variables, in that sense, we have a bivariate analysis.
Response 3.7.10: We used the univariate analysis (two-way analysis of variance) by SPSS. The dependent variable was the neutralizing antibody titers, the degree of severity, and underlying medical conditions were used as fixed factors and we clarified this point in the manuscript.
Comment 3.7.11: To use only Figure 1 or Table 2. Both presented the same data. Also, it is important to include the P-values and the statistical test used to perform the study.
Response 3.7.11: We thank the editor for this point and Table 2 has been excluded.
Figure 1.
Comment 3.7.12: Figure 1. To correct “moderate” to “Moderate”. Also, in my opinion, the authors should only cite intubated and non-intubated.
Response 3.7.12: The correction has been done in the figure 1.
Figure 2.
Comment 3.7.13: Part A is difficult to read. Part B. The authors can delete the number above the 1. The P-values should be cited. Also, the authors should classify the correlations between IMT and IGA, IGG, and IGM as moderate, for example.
Response 3.7.13: The figure files were uploaded to the Preflight Analysis and Conversion Engine (PACE) digital diagnostic tool, https://pacev2.apexcovantage.com/ to meet the requirements of the journal.
Figure 3.
Comment 3.7.14: Figure 3 is great. However, the authors should include a detailed legend.
Response 3.7.14: We thank the referee and a detailed legend has been included in the modified version.
Table 2
Comment 3.7.15: Table 2 can be excluded.
Response 3.7.15: Table 2 has been excluded as mentioned above.
Comment 3.7.16: Supplementary file 1 was not uploaded.
Response 3.7.16: we’re sorry for this inconvenience. According to the ethics committee board in Makkah, the data of patients cannot be shared in public and consequently we removed it also from the main manuscript. But we can share it for verification purposes.
Comments 3.8 Discussion:
No comments
Comments 3.9 Conclusion:
Comment 3.9: The authors can conclude the study considering its limitations. It is difficult to generalize the findings.
Response 3.9: We thank the referee and agree with him about the conclusion and this is mentioned in the manuscript.
Round 2
Reviewer 3 Report
The study was two main limitations: (i) a low number of individuals enrolled; and (ii) the study was restricted to only a region decreasing the generality of its findings.
This manuscript is a resubmission of an earlier submission. The following is a list of the peer review reports and author responses from that submission.
Round 1
Reviewer 1 Report
The paper describes the results of a complete serological test campaign for SARS-CoV-2 virus antibodies performed on a group of 50 patients vith preliminary positive swab test result and showing moderate or severe symptoms.
The data acquisition, their analysis and the result of the study are well described in the paper.
It is undoubt the usefulness of such data and related analysis. One weakness of the paper, underlined also by the authors, is related to the statistical sample considered: only infected individuals with evident symptoms; no asymptomatic or pre-simptomatic are included. However, the paper represents an interesting contribution in the advancement of our knowledge of the individual's reactions to the infection. The discussion could be enriched reporting explicitely the data of the papers used for comparison purposes so that the reader is facilitated.
Reviewer 2 Report
The paper by Alzahrani et al. presents results of SARS-CoV2 testing in the group of 50 PCR- confirmed hospitalized patients with COVID-19. IgG and IgM antibodies against SARS-CoV2 S-antigen were measured using in house ELISA and microneutralisation assays. Generally, the data obtained corresponds to the previously published results and possess no new original information, which would add to the present knowledge on a diagnosis or immunological response to SARS-CoV2 infection. The authors´ conclusion suggesting SARS-CoV 2 serology as complementary test to PCR for a diagnosis or the disease clearance may be misleading when high rate of anti-S antibodies false negativity in acute and long term persistence of in convalescent sera are considered and should be recommended only in special cases (see Hanson K et al., 2020; De Hartog G et al. 2021).
Comments:
SARS-CoV2 PCR positivity after clearance of COVID symptoms is caused by prolonged asymptomatic virus shedding and cannot be considered as the test false positivity. (Background, line 98).
Background, line 114: … a highly immunogenic protein (or immuno-dominant protein) …
Methods, ELISA: The data on validation of the in-house ELISA test used are not available. Will be the results obtained reproducible also using commercial ELISA test?
Statistical analysis: …. correlation between SARS-CoV2-specific antibodies measured by neutralisation and ELISA tests, respectively?
Results: Specificity of the ELISA tests used cannot be evaluated in the group of virologically confirmed COVID-19 patients. Possible cross-reactivity of anti-S IgG ELISA with other coronaviruses should be discussed (see Lv H et al., 2020).
Discussion – line 252: At present, exact correlates of protective immunity against SARS-CoV2 infection are still matter of debate (see Sadarangani et al., 2021).
Line 272: 80% positivity of anti SARS-CoV2 IgG reported here does not correspond 93% sensitivity declared in the Results (line 215).
Citation of the studies, that are in contradiction with the authors´ results, are missing. Moreover, the declaration on line 278 is in discrepancy with claiming on line 287.
Study limitations: To evaluate specificity of the tests used, sera from healthy individuals, not experiencing SARS-CoV2 infection, should be included, too.